# Probing coherent quantum thermodynamics using a trapped ion

O. Onishchenko[1,6], G. Guarnieri [2,3,6] ✉, P. Rosillo-Rodes[4], D. Pijn [1], J. Hilder [1], U. G. Poschinger[1], M. Perarnau-Llobet [5], J. Eisert [3] & F. Schmidt-Kaler [1]

Quantum thermodynamics is aimed at grasping thermodynamic laws as they apply to thermal machines operating in the deep quantum regime, where coherence and entanglement are expected to matter. Despite substantial progress, however, it has remained difficult to develop thermal machines in which such quantum effects are observed to be of pivotal importance. In this work, we demonstrate the possibility to experimentally measure and benchmark a genuine quantum correction, induced by quantum friction, to the classical work fluctuation-dissipation relation. This is achieved by combining laser-induced coherent Hamiltonian rotations and energy measurements on a trapped ion. Our results demonstrate that recent developments in stochastic quantum thermodynamics can be used to benchmark and unambiguously distinguish genuine quantum coherent signatures generated along driving protocols, even in presence of experimental SPAM errors and, most importantly, beyond the regimes for which theoretical predictions are available (e.g., in slow driving).

One of the pillars on which modern physics rests is classical phenomenological thermodynamics. Born out of the scrutiny of the functioning of heat engines, it is one of the most profound theories available, offering a wealth of applications. Its strength originates from the fact that it offers an effective description of complex systems in terms of a small number of macroscopic quantities. In recent years, it has become increasingly clear that the principles of thermodynamics must be sharpened in the quantum regime, where coherence, entanglement and quantum fluctuations play a significant role[1–5]. The prospect of harnessing these quantum features and exploiting them in order to outperform classical counterparts has boosted research across many fields, ranging from quantum biology[6] to quantum computation[7–10]. It also includes thermodynamics, where coherence has been shown to enhance the maximum cooling power or reach higher efficiencies[11–14]. While such development was unforeseeable for the founders of thermodynamics, the question of how thermodynamic notions should be altered has become highly relevant in light of the

progress in quantum engineering: the past few decades have seen huge leaps towards experimental realisations of meso- and nano-scale devices[15–20], culminating in emerging quantum technologies[21].

A rich body of theoretical work has provided guidelines for this exciting development in quantum thermodynamics[1–5,22]. However, to see evidence of genuine quantum effects in experimentally realised microscopic thermal machines seems to be harder to come across.

Thermal machines operating with single quantum systems and featuring signatures of quantum coherence have been devised[20,23–27] and compared to their classical counterparts[28]; but the task of actually verifying genuine quantum signatures that have no classical analogue in thermodynamics remains elusive. It has become clear that fluctuations, as inspired by the notions of quantum stochastic thermodynamics[29–32], maybe the tool that offers to discriminate quantum from classical prescriptions. Within this general framework, all thermodynamic quantities (such as work, heat, etc) become stochastic variables described by probability distributions, from which

[1]QUANTUM, Institut für Physik, Universität Mainz, Staudingerweg 7, 55128 Mainz, Germany. [2]Department of Physics and INFN - Sezione di Pavia, University of Pavia, Via Bassi 6, 27100 Pavia, Italy. [3]Dahlem Center for Complex Quantum Systems, Freie Universität Berlin, 14195 Berlin, Germany. [4]Institute for Cross-Disciplinary Physics and Complex Systems, Campus Universitat de les Illes Balears, E-07122 Palma, Spain. [5]Department of Applied Physics, University of Geneva, 1211 Geneva, Switzerland. [6]These authors contributed equally: O. Onishchenko, G. Guarnieri. ✉e-mail: giacomo.guarnieri@unipv.it

fluctuations can be computed[29,30,33]. Quantum effects most prominently manifest as such fluctuations[31,32,34–43]. These have a stark impact on the work fluctuation-dissipation relation[44–46] or on thermodynamic uncertainty relations[33,47–50]. In its original formulation, the FDR expresses a direct proportionality between the average dissipated work and its equilibrium fluctuations

$$\frac{\beta}{2}\text{Var}(W) = \langle W \rangle - \Delta F, \tag{1}$$

and it is valid for processes where the system remains close to thermal equilibrium at all times[51]. Specifically, Eq. (1) relates the first two cumulants of the work distribution, i.e., $\langle W \rangle$ and Var($W$), for a slowly driven system in contact with a thermal bath at inverse temperature $\beta > 0$, with $\Delta F$ being the change in free energy between the two endpoints of the process. Eq. (1) has moreover been confirmed in various experimental platforms in mesoscopic systems[52,53].

However, it has recently been demonstrated that Eq. (1) is violated in the presence of quantum friction[54,55], i.e. the generation of quantum coherence in the instantaneous energy eigenbasis[46,54,55] due to an external Hamiltonian driving such that $[H(t),H(t')] \neq 0$ for $t' \neq t$. This result relies on two assumptions: (i) initialisation of the system in a thermal (i.e., Gibbs) equilibrium state and (ii) a slow-driving protocol so that the system remains close to equilibrium at all times.

Fast and far-from-equilibrium processes are of great utility to real-world applications of quantum devices, such as e.g. in computing, where shorter operation times are indeed desirable. This regime is still how an almost uncharted territory for stochastic thermodynamics, however. In this work, we report on the experimental analysis of a genuine quantum correction using a single trapped-ion qubit beyond (ii), thus providing evidence that the quantum-friction induced violation to the work FDR Eq. (1) theoretically demonstrated in slow-driving regime is valid and measurable even beyond this working assumption. Our results clearly show that such correction measurements are incompatible by more than 10.9$\sigma$ with values obtained from any incoherent (i.e., classical) protocol at finite driving speed and by more than 12.1$\sigma$ with values stemming from state preparation and measurement (SPAM) errors. The core point is that we are not making a prediction and are testing whether the prediction is approximately valid in the experiment. Instead, reminiscent of a test of a Bell inequality, we take data, and ask to what extent these data are compatible with an incoherent evolution. This conclusively showcases the potential of quantum stochastic thermodynamics in order to certify a genuine quantum effect.

## Results

### Quantum fluctuation-dissipation relation protocol

A viable protocol for testing the measurability of quantum violations to the work FDR with a single qubit consists of dynamically changing the Hamiltonian according to

$$\hat{H}(\theta_t) = \frac{\hbar\omega_q}{2}\left(\sin(\theta_t)\hat{\sigma}_y - \cos(\theta_t)\hat{\sigma}_z\right), \tag{2}$$

by varying the angle parameter $\theta_t$ from 0 to $\pi/2$. In the following, we set $\hbar\omega_q = 1$, i.e., all energy quantities are rescaled to the qubit frequency $\omega_q$. The Hamiltonian Eq. (2) does not commute with itself at different times, i.e., for different values of $\theta_t$. The qubit has to be kept close to thermal equilibrium with respect to the instantaneous Hamiltonian, which requires coupling the qubit to a thermal bath and a sufficiently slow drive process. As discussed in ref. 55, this slow continuous protocol can be replaced by a step-wise protocol, where thermalisation processes are combined with coherent rotations induced by driving (2). Such a discrete protocol, which is more convenient for our experimental setup, allows for observation of the violation of the FDR

for finite values of the step number $N$ with an error of order $\mathcal{O}(1/N^2)$ when compared to continuous protocols.

Here we consider a discrete process in which the parameter $\theta_t$ in (2) is varied from $\theta_0 = 0$ to $\theta_N = \pi/2$ in $N$ equal steps of $\Delta\theta = \pi/(2N)$. For the $j$th step, with $\theta_j := j\Delta\theta$, the following sequence of operations are carried out, as depicted in Fig. 1:

(i) The state is prepared in a thermal state of $\hat{H}(\theta_j)$, i.e., $\rho_j = e^{-\beta\hat{H}(\theta_j)}/\mathscr{Z}$ with the partition function $\mathscr{Z} = 1 + e^{-\beta}$. For $j = 0$, it reads

$$\hat{\rho}_0 = \frac{1}{\mathscr{Z}}\left(|0\rangle\langle 0| + e^{-\beta}|1\rangle\langle 1|\right), \tag{3}$$

with $\beta > 0$ being related to the population $p$ of $|1\rangle$ via[25]

$$\text{Tr}(\hat{\sigma}_z\hat{\rho}) = 1 - 2p = \tanh\left(\frac{\beta}{2}\right). \tag{4}$$

(ii) A projective energy measurement is performed on $\rho_j$ in the basis $\hat{H}(\theta_j)$, obtaining outcome $e_j \in \{0, 1\}$.

(iii) The post-measurement state is coherently evolved according to

$$\hat{R}(\Delta\theta) = \exp\left(-i\frac{\Delta\theta}{2}\hat{\sigma}_x\right). \tag{5}$$

(iv) A second projective measurement of $\hat{H}(\theta_j)$ is performed, yielding outcome $e'$. Using that $\hat{H}(\theta_{j+1}) = R(\Delta\theta)H(\theta_j)R^\dagger(\Delta\theta)$, we note that steps (iii)–(iv) are equivalent to a Hamiltonian quench $\hat{H}(\theta_j) \mapsto \hat{H}(\theta_{j+1})$ followed by a projective measurement of $\hat{H}(\theta_{j+1})$, as originally considered in ref. 55.

According to the standard two-point measurement (TPM) scheme[56–58], the difference in these two energy measurement

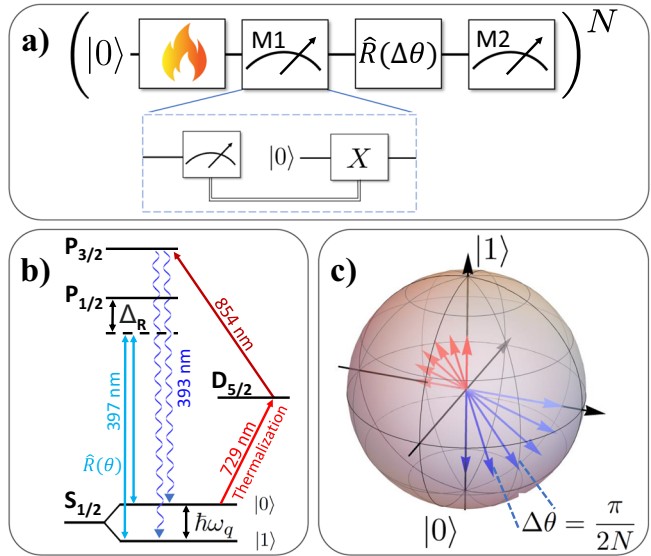

**Fig. 1 | Experimental implementation scheme for detecting quantum work fluctuations using a trapped-ion qubit. a** Sequence of operations for obtaining a single measured value of work. The flame represents thermalisation to a Gibbs state in the computational basis. $|0\rangle$ represents qubit initialisation. A two-point measurement is realised around a work step, given by a qubit rotation $\hat{R}(\theta)$. The inset shows a non-demolition energy measurement, which is emulated via conditional re-initialisation (see text). **b** The relevant energy levels of a $^{40}$Ca$^+$ ion. The qubit is encoded in the Zeeman sub-levels of the $4\,^2S_{1/2}$ ground state. **c** Discrete protocol visualised on the Bloch sphere, with the arrows showing the instantaneous eigenstates of the Hamiltonian Eq. (2). In the step-wise protocol, these are separated by the fixed angle $\Delta\theta$. The low (high) energy contributions to the thermal Gibbs state Eq. (3) are shown in blue (red).

outcomes allows us to define the work performed on the system along step $j$: $w_j = e' - e_j$[56-58], which therefore becomes a random variable distributed according to

$$\Pr(w_j) = \sum_{e_j, e'_j = 0,1} \delta[w_j - (e'_j - e_j)] \langle e_j | \hat{\rho}_j | e_j \rangle |\langle e_j | e'_j \rangle|^2. \quad (6)$$

This represents a proper definition since, during each quench, the system is a closed externally driven system so its change in internal energy can be only interpreted as work. Explicitly, since our system is given by a qubit, one has that $w_j = \{-1, 0, 1\}$ for the coherent protocol under consideration with probabilities

$$\Pr(w_j = +1) = (1 - p) \sin^2(\Delta\theta/2), \quad (7)$$

$$\Pr(w_j = -1) = p \sin^2(\Delta\theta/2), \quad (8)$$

$$\Pr(w_j = 0) = 1 - \Pr(w_j = +1) - \Pr(w_j = -1). \quad (9)$$

Finally, the total work done along the protocol is the sum of the step work values: $W = \sum_{j=0}^{N-1} w_j$, which is a stochastic quantity because of thermal and quantum fluctuations. Since each thermalisation step resets any information that has been available in the previous state, the protocol is effectively Markovian, and the step work probabilities Eq. (9) are statistically identical and independent. This justifies starting each step in the state (3) for the sake of experimental simplicity. The above experimental protocol is illustrated in Fig. 1.

From the first and second cumulants of the work distribution, i.e., its mean $\langle W \rangle$ and variance $\mathrm{Var}(W)$, we quantify the violation of Eq. (1) by the difference

$$\mathscr{Q} = \frac{\beta}{2} \mathrm{Var}(W) - [\langle W \rangle - \Delta F]. \quad (10)$$

Note that for the processes considered, $\Delta F = 0$ as the free energy is invariant under a basis change given by the effective Hamiltonian Eq. (2). Explicit analytical expressions for the quantities appearing in Eq. (10) are provided in Supplementary Notes for the derivation or work distribution cumulants, an error estimation discussion and experimental implementation details.

## Experimental implementation

The qubit is experimentally encoded in the spin of the valence electron of a $^{40}$Ca$^+$ ion[59] confined in a segmented Paul trap[60]. Figure 1b shows the relevant energy levels and transitions. A static magnetic field gives rise to a frequency splitting of $\omega_q \approx 2\pi \times 10.5$ MHz between the Zeeman sub-levels of the $4^2$S$_{1/2}$ ground state, which are taken to be the logical basis vectors $|0\rangle := |m_j = +1/2\rangle$ and $|1\rangle := |m_j = -1/2\rangle$[61]. Qubit initialisation to state vector $|0\rangle$ is realised via optical pumping, i.e., elective depletion of $|1\rangle$, see Fig. 1b and (Supplementary Notes for the derivation or work distribution cumulants, an error estimation discussion, and experimental implementation details). A (thermal) Gibbs state in the logical basis $\{|0\rangle, |1\rangle\}$ is prepared by partial population transfer from $|0\rangle$ to $|1\rangle$ in conjunction with the first qubit measurement of the TPM: for a given value of $\beta$, a fixed amount of population is transferred to the meta-stable state, and the first measurement leads to a projection to the logical basis vector $|0\rangle, |1\rangle$ with the Boltzmann weights according to Eq. (4), see (Supplementary Notes for the derivation or work distribution cumulants, an error estimation discussion, and experimental implementation details).

Coherent qubit rotations are performed via stimulated Raman transitions driven by a pair of co-propagating off-resonant beams, far red-detuned from the $4^2$S$_{1/2} \leftrightarrow 4^2$P$_{1/2}$ transition by $\Delta_R \approx 2\pi \times 250$ GHz. With a difference frequency between the beams matched to the qubit

frequency, this realises a resonant qubit drive, insensitive to the motional state of the ion. Switching the beams on for a defined exposure time generates the evolution $\hat{R}(\Delta\theta) = \exp(-i(\Delta\theta/2)\hat{\sigma}_x)$, with the rotation angle (pulse area) $\Delta\theta$ being determined by the beam intensities and the exposure time. Qubit readout is performed by selective population transfer from $|0\rangle$ to the meta-stable state $3^2$D$_{5/2}$[59]. After that, the detection of state-dependent fluorescence using 397-nm light reveals the result $|1\rangle$ ("bright") or $|0\rangle$ ("dark"). This qubit readout is destructive, as the post-measurement state ends up being completely depolarised. Realising a projection-valued measurement therefore requires re-initialising the qubit after a measurement by optical pumping, followed by a $\pi$-pulse conditional on the previous measurement result, see Fig. 1a.

For a given choice of the parameters $N$ and $\beta$, we collect *work samples* by performing $N$ independent runs of the sequence and storing the step work values $w_j$, assigning $w_j = +1(-1)$ if the first readout result is $|0\rangle(|1\rangle)$ and the second result is $|1\rangle(|0\rangle)$ and $w_j = 0$ otherwise.

We first characterise the quantum correction $\mathscr{Q}$ to the FDR by collecting work samples for different values of the step number $N$, ranging from $N = 2$ to $N = 7$, at a fixed inverse temperature $\beta = 3.413 \pm 0.025$, corresponding to an excited state population $p = 0.032 \pm 0.001$. For each value of $N$, we repeat the protocol 8000 times and compute the sample mean and sample variance from the total work values $W$ pertaining to each work sample. This allows us to reveal the quantum correction Eq. (10) for different values of $N$, see Fig. 2. Note that the observed excess fluctuations systematically fall short of the expected ideal values as the rotation pulse calibration is prone to systemic effects induced by measurement errors. To quantitatively certify that this value of $\mathscr{Q}$ is a genuine quantum effect and not one caused by thermal fluctuations, finite $N$, or experimental imperfections, we compare the results to computed $\mathscr{Q}$ values, which would be obtained from an incoherent protocol or from measurement errors. We point out that, since the time taken by the protocol detailed above is much shorter than the coherence time[62], we can consider the system as isolated for the whole duration of the experiment. Let us first introduce the notion of speed as $v = \|\Delta H\|/N$, where $\|\Delta H\|$ denotes the operator norm of the change in the system's Hamiltonian and where the slow-driving regime is recovered when $v \ll 1$. We then proceed by simulating hundreds of thousands of incoherent protocols, where the Hamiltonian is varied such that it still commutes with itself at all times. Such protocols consist of changing the qubit frequency from $\omega_q^{(i)}$ to $\omega_q^{(f)}$ in $N$ discrete steps, while keeping the energy eigenbasis fixed to $\hat{\sigma}_z$. For such protocols, $\|\Delta H\| = (\omega_q^{(f)} - \omega_q^{(0)})/2$, while for the coherent protocol performed in the experiment, one has $\|\Delta H\| = 1/\sqrt{2}$. In Supplementary Note 1 of the Supplementary Information file, we have provided all the details about incoherent processes. In order to quantitatively compare our experimental observations with the values of FDR corrections compatible with incoherent processes for any possible driving speed, we plot the rescaled quantity $N\mathscr{Q}/\|\Delta H\| \equiv \mathscr{Q}/v$. The results of these simulations give rise to a bounding region for $\mathscr{Q}$ values observed for purely incoherent driving, which is shown in Fig. 2, together with the experimentally measured values of $N\mathscr{Q}/\|\Delta H\|$ and the corresponding fully coherent driving protocol. We see that for incoherent processes, $\mathscr{Q}/v \propto 1/N$, which indicates that Eq. (1) is valid in the slow-driving regime. By contrast, quantum processes $\mathscr{Q}/v$ tends to a constant, and our experimental points lie beyond the region of values for $N\mathscr{Q}/\|\Delta H\|$ attainable by any incoherent process. This provides a piece of striking evidence that those measured corrections are only compatible with a genuinely quantum coherent process.

In contrast to the ideal case where energy measurements in the TPM scheme are error-free, experimental measurement-readout errors may occur. In our setup, the second measurement of each TPM has a small but non-zero conditional probability $p_{d|1}$ of incorrectly reading out the qubit as "dark" when it was in the 'bright" state vector $|1\rangle$,

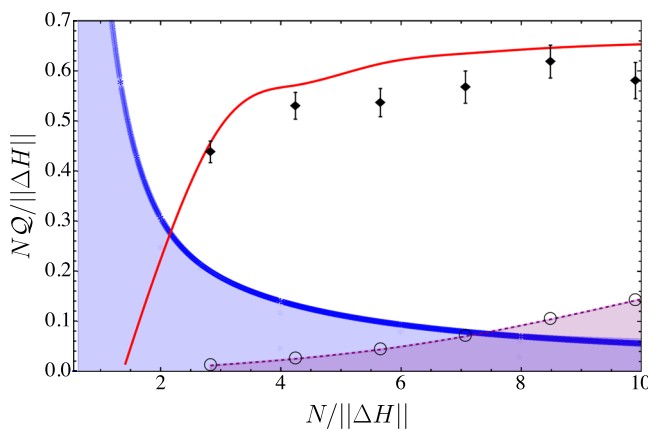

**Fig. 2 | Quantum correction $N\mathcal{Q}/\parallel\Delta H\parallel$ as a function of the inverse of the process velocity $v^{-1} = N/\parallel\Delta H\parallel$.** The slow-driving regime $v \ll 1$ corresponds to the large number of subdivisions $N \gg 1$. The results of the experimental measurements (black diamonds) are shown together with the theoretical curve (solid red line) calculated for a discrete fully coherent protocol. The plot also displays the simulations of incoherent processes (blue region) obtained different dynamically varied qubit energy gaps $\omega_f - \omega_0 = 2\parallel\Delta H\parallel$ and subdivisions numbers $N$. Finally, values of $N\mathcal{Q}_{SPAM}/\parallel\Delta H\parallel$ which would arise purely from measurement errors are also displayed (black empty circle marker). This plot shows that the experimental points have a finite separation from the region of incoherent processes (blue) and from the SPAM error region (light-purple), thus quantitatively proving that the measured quantum correction $N\mathcal{Q}/\parallel\Delta H\parallel$ is a genuine quantum signature.

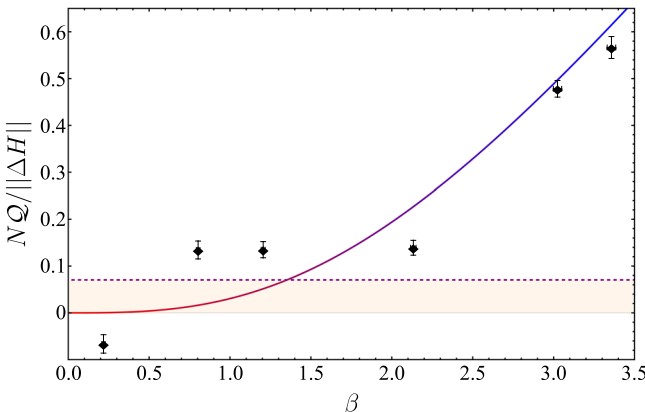

**Fig. 3 | Measured quantum correction $N\mathcal{Q}/\parallel\Delta H\parallel$ as a function of the inverse temperature for $N = 5$.** Error bars are due to the counting statistics with 8000 repetitions. The theory prediction Eq. (11) is plotted from high (red) to low (blue) temperatures, without free parameters. The maximum SPAM-induced fluctuation readout is denoted by the dashed line.

Figs. 2 and 3 are independent of the relevant energy scales of the experiment and, therefore, agnostic to the employed platform.

## Discussion

In this work, we have exploited the excellent degree of control offered by a trapped-ion qubit platform to perform the first measurement and unambiguous detection of a genuinely quantum thermodynamic signature, namely the quantum correction to the work fluctuation-dissipation relation. This has been realized by performing a sequence of $N$ alternating coherent drives, thermalization steps and energy non-demolition measurements on a single qubit. Our result has revealed a quantum correction to the classical work FDR Eq. (1), which has been proven to be statistically incompatible by >10.9 standard deviations with any incoherent protocol, and incompatible by more than 12.1 standard deviations with any SPAM-induced error. This conclusion thus certifies the genuine quantum nature of our measurements, even beyond the slow-driving regime. Moreover, we have shown that the spurious correction to the FDR induced by small, but non-zero, error probabilities in the measurement readout, which we called $N\mathcal{Q}_{SPAM}/\parallel\Delta H\parallel$, has the general property of linearly growing with the number of subdivisions $N$, a scaling in stark contrast both with the incoherent $N\mathcal{Q}_{inco}/\parallel\Delta H\parallel$, which decreases as $1/N$, and with the quantum correction $N\mathcal{Q}/\parallel\Delta H\parallel$, which approaches a constant positive asymptote.

We believe that our research represents a significant result in the direction of experimental observation and certification of genuine quantum effects on small-scale platforms. This invites further exciting endeavours: for example, observing quantum effects in quantum field thermal machines in the realm of quantum many-body physics[63]. It also seems conceivable that different quantum corrections can be measured in work extraction experiments, witnessing temporal coherence reflecting non-Markovian quantum dynamics[64–66], a feature that could be exploited in order to further curb the SPAM measurement-readout errors, allowing to access slower protocols at higher $N$. One may also bring these experimental results into contact with a body of theoretical work on single-shot work extraction[22,67,68]. We expect this work to stimulate further efforts of experimentally exploring the deep quantum regime in quantum thermodynamics.

## Data availability

The experimental data generated during the current study will be made available upon request to the corresponding author.

and vice-versa. As we show in detail in Supplementary Note 2 of the Supplementary Information file, the introduction of these measurement-readout errors leads to a spurious non-zero correction $N\mathcal{Q}_{SPAM}/\parallel\Delta H\parallel$. Figure 2 shows the region of values of $N\mathcal{Q}/\parallel\Delta H\parallel$ that would be indistinguishable from a worst-case scenario $N\mathcal{Q}_{SPAM}/\parallel\Delta H\parallel$ computed by performing energy measurements without any in-between qubit rotations. Our result shows that the experimental points have a statistical distance above $12.1\sigma$ both to the regions of values of $N\mathcal{Q}$ compatible with incoherent processes and with SPAM errors.

While the quantum coherent excess work fluctuations $N\mathcal{Q}/\parallel\Delta H\parallel$ asymptotically settle to a constant value for increasing $N$, the spurious contribution from SPAM errors $N\mathcal{Q}_{SPAM}/\parallel\Delta H\parallel$ increases with $N$. This has the important consequence that the observation of the genuine quantum correction Eq. (10) would be completely hindered by SPAM errors in the quasi-static limit $N \rightarrow \infty$. It is precisely the outstanding control offered by trapped-ion platforms that allows for comparatively small readout error probabilities, enabling the witness of a quantum correction to the classical FDR in the regime of intermediate driving speeds.

We complete our analysis by measuring $N\mathcal{Q}$ as a function of the temperature for a fixed number of subdivisions $N = 5$, see Fig. 3. First of all, it can be clearly seen that the quantum correction $N\mathcal{Q}$ correctly reproduces the behaviour

$$N\mathcal{Q}/\parallel\Delta H\parallel = (\pi^2\sqrt{2}/4)\left[\beta/2 - \tanh(\beta/2)\right] + \mathcal{O}(1/N)$$
$$\simeq (\pi^2\sqrt{2}/4)\left[0 + \mathcal{O}(\beta^2)\right] + \mathcal{O}(1/N), \quad (11)$$

where the second line shows a quadratic decrease to zero in the high-temperature limit, where thermal fluctuations dominate. At low temperatures, the excess fluctuations $\mathcal{Q}$ emerge from quantum coherence. For high temperatures, the measured $\mathcal{Q}$ display excess deviations from the expected values. We attribute this to drifts in the parameters of the thermalisation step, which is increasingly prone to errors for higher spin temperatures. Note that the results shown in

## Code availability

The code written to analyse data for this work will be made available upon request to the corresponding author.

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

## Acknowledgements
We warmly thank J. Anders for helpful comments, and H. Miller and M. Scandi for insightful discussions. This work has been funded by the DFG (FOR 2724, for which this is an inter-node collaboration reaching an important milestone, and CRC 183), the FQXI, the ERC (DebuQC) and the Quantum Flagship (Millenion, for which is again the result of an inter-node collaboration), the BMBF (DAQC). G.G. acknowledges funding from the European Union's Horizon 2020 research and innovation programme under the Marie Sklodowska-Curie Grant Agreement INTREPID, no. 101026667. M.P.L. acknowledges funding from the Swiss National Science Foundation through an Ambizione Grant no. PZ00P2-186067.

## Author contributions
O.O. and G.G. contributed equally to this work. G.G., M.P.-L., P.R and J.E. developed the theory, while O.O., D.P., J.H., U.G.P. and F.S.-K. performed the experiment and collected the data. G.G., U.G.P. and O.O. wrote the code to analyse the data and produce the plots. All the authors contributed to writing the manuscript.

## Funding

## Competing interests
The authors declare no competing interests.
