## [Peer Review File · Nature Communications]

Probing coherent quantum thermodynamics using a trapped ionREVIEWER COMMENTS

Reviewer #1 (Remarks to the Author):

The manuscript describes an experimental demonstration of the violation of the work fluctuation-dissipation relation (FDR) with a single trapped ion qubit. Authors perform two measurements of the qubit energy before and after the work is applied to the qubit, calculate mean and variance of the work, and observe violation of the work FDR. The paper is well written, the experimental procedure and data analysis is clearly explained. However, in my opinion, some details of the work require further clarification before I can recommend the manuscript for publication.

In particular:

1. The question “if there are violations of the work fluctuation-dissipation relation” looks like a question that can be resolved by a theory alone. If we do not expect that a single qubit system deviates from the law of quantum mechanics, there is no reason to believe that calculations in the section 1 of the supplemental information will deviate from experiment. These calculations clearly show that quantum mechanics predicts violation of the work fluctuation-dissipation relation. The experiment simply confirms that authors have good enough control of a single qubit to observe predictions of quantum mechanics, which is not surprising given high single qubit fidelities routinely achieved in modern quantum computing experiments. What is the value of doing the experiment in this case?
2. The first state preparation and measurement M1 on Fig 1a effectively implements a quantum random number generator that randomly prepares qubit in state $|0\rangle$ or $|1\rangle$. Can this step be replaced by a random state preparation controlled by a classical computer? It seems to me that such replacement will not change any experimental observables but will greatly simplify experimental protocol. The same “Markovian dynamics” argument used in the manuscript would be applicable to justify it.

3. Authors mentioned in the manuscript that the qubit rotations are performed by a pair of Raman beams. Are these beams co-propagating? If not, what is the beam geometry and what is the ion temperature? My concern is that non-copropating Raman beams can couple spin and motional states of the ions even when they drive carrier transition, especially if the ion temperature is high. In such a scenario a simple one qubit model cannot be used to analyze the experiment. Hope authors can clarify it.

4. The authors define work as the difference of the two energy measurement outcomes. Such a definition might be obvious to the experts in quantum thermodynamics, but it would be helpful for a more general audience of Nature Communications to explain how it is related to a classical definition of work as "energy transferred to an object via the application of force along a displacement". What is the object here? What kind of force is acting on it, and why the energy transfer should be measured as described in the manuscript?

5. It is not very clear what data is presented in Table 1. After a long and detailed study of the manuscript I was able to guess that the points in column "Point" correspond to the data points from Figure 2 (is it correct?). However this is not explained neither in the table caption nor in the text of the manuscript. In addition, why do authors choose to express Δ^{SPAM} and Δ^{inc} in rows 3-6 as the function of σ_3 ?

6. Either experimental data on Fig 3 do not match the theory plot or the error bars on Fig 3 are severely underestimated. Only one out of 7 error bars cross the theory plot, which is statistically unlikely. It looks like the experiment is suffering from some systematic errors that are not fully taken into account. The manuscript would benefit from a more detailed explanation of this discrepancy.

In conclusion, the manuscript needs further clarifications before I will be able to recommend it for publication in Nature Communications.

Reviewer #2 (Remarks to the Author):

This paper reports on the realization of an experimental measurement of a genuine quantum correction to the classical fluctuation-dissipation relation (FDR) using a single trapped ion. The measurement of the quantum correction to the fluctuation-dissipation relation is done by performing a sequence of coherent drivings and measurements of the state of the ion. The results show agreement with the recently proven quantum work FDR, violating the classical FDR by more than ten standard deviations. The paper also discusses how statistical and systematic errors are estimated, and how parameter drifts can affect the estimation of spurious corrections.

As explained in the paper this is the experimental implementation of the theoretical ideas developed mainly in Ref. [43,44]. There it is shown that the classical FDR is violated in the presence of so-called quantum friction. This refers to the dynamical situation in which the system develops coherences in the instantaneous energy eigenbasis during the evolution. To observe such deviation it is necessary to implement a dynamics in which the state of the system can develop coherences and further coupling it to a bath that allows a thermalization in the instantaneous energy eigenbasis. However, as shown, since the work statistics is evaluated via the usual two-point measurement approach, each of these measurements are identical and independent. As a result, there is no need for thermalization steps in the actual implementation. Thus, the experimental results are collected by simply considering independent stages that require: the preparation of different initial states of the two level system, undergoing some unitary evolution and ultimately carrying out a projective measurement of its state.

The article is well written, and the main issues touched upon are timely and of interest to a broad scientific community. However, given that the main contribution of the paper is the experimental confirmation of a quantum correction proposed in Ref. [43,44], my only concern is related to the relevance of the experiment that is presented. After reading the manuscript my impression is that this seems a standard procedure in this particular experimental platform, but there could be some crucial details or elements which I've overlooked or missed entirely. Therefore, I am willing to accept the article for publication, provided the authors present a compelling argument as to why this should be published in

such a high-impact journal.

Reviewer #3 (Remarks to the Author):

The authors describe an experimental scheme of a trapped ion qubit to verify the quantum work fluctuation-dissipation relation. They performed a sequence of two-time work measurements with a coherent rotational operation in between the measurements. The state preparation and measurement errors are used to deduce the standard deviation, the correction to the work fluctuation-dissipation relation. The paper is well written but would benefit from recapping the derivation of the fluctuation relation. Although the experimental study of fluctuation-dissipation is interesting to the on-going effort to the nonequilibrium phenomena in the information quantum thermodynamics, the present study appears to miss the measurement contribution part. In addition, it is not clear how the violation of the fluctuation-dissipation relation at high temperature is related to the 'genuine' quantum correction. And how does the study differ from other trapped-ion studies on the verification of Jarzynski equality and Landauer principles. In summary, I cannot recommend the article for publication in Nature Communication in its current form.

Reply to Reviewer #1

The manuscript describes an experimental demonstration of the violation of the work fluctuation-dissipation relation (FDR) with a single trapped ion qubit. Authors perform two measurements of the qubit energy before and after the work is applied to the qubit, calculate mean and variance of the work, and observe violation of the work FDR. The paper is well written, the experimental procedure and data analysis is clearly explained.

We would like to thank the referee for this helpful and positive report. We are delighted to read that the reviewer thinks that our work is “well written, the experimental procedure and data analysis is clearly explained”. The reviewer suggests a number of changes that reflect clarifications from our side. We have carefully adapted our manuscript accordingly. Below, we explain in point-to-point replies what we have done to accommodate the suggestions.

However, in my opinion, some details of the work require further clarification before I can recommend the manuscript for publication.

We are thankful for this.

In particular:

1. The question “if there are violations of the work fluctuation-dissipation relation” looks like a question that can be resolved by a theory alone. If we do not expect that a single qubit system deviates from the law of quantum mechanics, there is no reason to believe that calculations in the section 1 of the supplemental information will deviate from experiment. These calculations clearly show that quantum mechanics predicts violation of the work fluctuation-dissipation relation. The experiment simply confirms that authors have good enough control of a single qubit to observe predictions of quantum mechanics, which is not surprising given high single qubit fidelities routinely achieved in modern quantum computing experiments. What is the value of doing the experiment in this case?

We thank the referee for giving us the opportunity to better explain and clarify the purpose and added value of this work and the reason why it significantly goes beyond existing work.

First of all, we would like to make an important general point that the purpose of our work is not to answer the question “whether there are violations of the work fluctuation-dissipation relation” (FDR from now on, for brevity), but to demonstrate the role and possibilities of non-equilibrium thermodynamics in unambiguously certifying genuine quantum signatures by showcasing the actual experimental measurability of FDR violations (once practical constraints and errors are included). Although this might not seem like it at first sight, we firmly believe that this point is crucial.

Even more importantly, all the theoretical predictions regarding the quantum violations of the FDR have insofar been derived and proven only within the slow driving regime. In the language of protocols consisting of N alternating steps of Hamiltonian quenches and thermalizations, such as in Ref. [44], the regime of slow-driving corresponds to the limit $N \rightarrow \infty$. According to the protocol we have considered in our work and implemented in the laboratory (described above Eq.(3)), the quenches consisted of qubit rotations of angles $\theta = 2\pi/N$. Experimentally accessing the slow-driving

regime would therefore require being able to perform coherent precise rotations of infinitesimal angles which, however, is currently still out of reach even for state-of-the-art platforms like ours despite the high single qubit fidelities and precision in qubit manipulations.

This practical constraint led us to experimentally investigate the stochastic thermodynamic landscape **beyond** the slow-driving regime, namely for finite and relatively small values of N (i.e. corresponding to finite-time and fast driving protocols), where **no theoretical predictions have been made yet**. The results of our work, therefore, provide valuable information for future advancements in the still uncharted territory of quantum work FDR for finite-time out-of-equilibrium processes.

Last but not least, continuing along the same argument, the actual physical constraints of our trapped-ion experiment led us to the characterization, for the first time, of the impact of SPAM errors on the measurement of the out-of-equilibrium work statistics, ultimately leading to the discovery of a purely SPAM-generated violation of the FDR that had not been even theoretically studied before.

These last two facts are precisely the reasons why in our work we have performed an in-depth statistical analysis aimed at proving that the measured violations of the work FDR were statistically incompatible with both a classical (i.e., incoherent) external driving at finite speed and with SPAM errors.

To summarize, the added value of our work lies in the fact that

(a) by making use of recent theoretical developments in the field of quantum stochastic thermodynamics, we have demonstrated how these results can be used to provide measurable and certifiable quantum signatures in a trapped-ion experiment. Furthermore and remarkably, this was achieved by only relying on simple energy measurements in the z basis of a qubit.

(b) by making use of the experiment and taking into account its physical constraints, we have investigated the violations of the fluctuation-dissipation relation beyond the predictions of current theoretical state-of-the-art, i.e., outside the slow-driving regime.

Thanks to the referee's comment, we have realized that these aspects were not sufficiently emphasized enough in the previous version of the manuscript, and we have now implemented significant changes in the revised version (see text marked in color throughout the paper) in order to fully clarify them.

2. The first state preparation and measurement M1 on Fig 1a effectively implements a quantum random number generator that randomly prepares qubit in state $|0\rangle$ or $|1\rangle$. Can this step be replaced by a random state preparation controlled by a classical computer? It seems to me that such replacement will not change any experimental observables but will greatly simplify experimental protocol. The same "Markovian dynamics" argument used in the manuscript would be applicable to justify it.

The referee is correct on the observation that the first state preparation and measurement effectively implements classical a random number generator. Replacing the laser-induced thermalization of the qubit with a computer-controlled random state preparation would not

provide a significant simplification of the experiment - the laser beam for optical pumping is required anyway, and there is no need to precisely adjust for a given spin temperature.

3. Authors mentioned in the manuscript that the qubit rotations are performed by a pair of Raman beams. Are these beams co-propagating? If not, what is the beam geometry and what is the ion temperature? My concern is that non-copropagating Raman beams can couple spin and motional states of the ions even when they drive carrier transition, especially if the ion temperature is high. In such a scenario a simple one qubit model cannot be used to analyze the experiment. Hope authors can clarify it.

The Raman beams are co-propagating and the qubit drive is insensitive to the ion motion. We now clarify this in the manuscript on page 3, left column, 3rd paragraph.

4. The authors define work as the difference of the two energy measurement outcomes. Such a definition might be obvious to the experts in quantum thermodynamics, but it would be helpful for a more general audience of Nature Communications to explain how it is related to a classical definition of work as "energy transferred to an object via the application of force along a displacement". What is the object here? What kind of force is acting on it, and why the energy transfer should be measured as described in the manuscript?

The object is the spin of the valence electron of the trapped ion, i.e. the qubit. The 'force' acting on the spin is the qubit drive. This is pointed out in the description of the protocol under 'quantum fluctuation-dissipation relation protocol' on page 2. The notions 'force' and 'displacement' pertain to continuous degrees of freedom - the most general definition of work holding for any kind of physical system means energy transfer caused by the change of energy eigenvalues induced by external controls, without redistribution of population. This is captured by the parametrized qubit Hamiltonian Eq. (2). Then, as discussed below Eq. (2), the continuous protocol for changing the Hamiltonian parameter θ can be replaced by a step-wise protocol, where jumps of θ are equivalent to qubit drive operations.

We stress that these definitions are in complete agreement with the standard "two-point measurement scheme" (TPM) to define work in driven quantum systems. As the referee has recommended, we have now added a thorough explanation of this scheme and the consequent definition of work; moreover, we have restructured the presentation of the protocol in order to further clarify these points.

5. It is not very clear what data is presented in Table 1. After a long and detailed study of the manuscript I was able to guess that the points in column "Point" correspond to the data points from Figure 2 (is it correct?). However this is not explained neither in the table caption nor in the text of the manuscript. In addition, why do authors choose to express Δ_j and Δ_j^{inc} in rows 3-6 as the function of σ_j ?

We, once more, thank the referee for pointing this out, and we agree that the previous table both contained some typos (in the rows 3-6, each Δ_j was meant and given in units of σ_j but was a typographic refuse in the subscript of σ_j) and was not clear. In the revised version of the paper, for the sake of readers' clarity we have decided to move the relevant (typos-fixed) information that was contained in that table either in the figure caption or in the main text.

6. Either experimental data on Fig 3 do not match the theory plot or the error bars on Fig 3 are severely underestimated. Only one out of 7 error bars cross the theory plot, which is statistically unlikely. It looks like the experiment is suffering from some systematic errors that are not fully taken into account. The manuscript would benefit from a more detailed explanation of this discrepancy.

The referee still correctly points out that the data is affected by residual systematic errors. These are due to the small-angle qubit rotations, which are notoriously hard to calibrate and strongly prone to laser intensity drifts. While we have taken as much care as possible in these calibrations, it is quite reasonable to suspect that we have not put all systematic effects under control. However, we do not report on a precision measurement in this work, but rather show a scenario where quantum physics renders the classical FDR to be invalid. This finding is fully supported by the shown data, irrespective of systematic errors of the measurement. Please see the corresponding statement in the manuscript on page 3, right column, first paragraph.

In conclusion, the manuscript needs further clarifications before I will be able to recommend it for publication in Nature Communications.

We thank once again the referee for their comments. Having carefully accommodated all comments, we hope that our work is now suitable for publication in Nature Communications.

Reply to Reviewer #2

This paper reports on the realization of an experimental measurement of a genuine quantum correction to the classical fluctuation-dissipation relation (FDR) using a single trapped ion. The measurement of the quantum correction to the fluctuation-dissipation relation is done by performing a sequence of coherent drivings and measurements of the state of the ion. The results show agreement with the recently proven quantum work FDR, violating the classical FDR by more than ten standard deviations. The paper also discusses how statistical and systematic errors are estimated, and how parameter drifts can affect the estimation of spurious corrections. As explained in the paper this is the experimental implementation of the theoretical ideas developed mainly in Ref. [43,44]. There it is shown that the classical FDR is violated in the presence of so-called quantum friction. This refers to the dynamical situation in which the system develops coherences in the instantaneous energy eigenbasis during the evolution. To observe such deviation it is necessary to implement a dynamics in which the state of the system can develop coherences and further coupling it to a bath that allows a thermalization in the instantaneous energy eigenbasis. However, as shown, since the work statistics is evaluated via the usual two-point measurement approach, each of these measurements are identical and independent. As a result, there is no need for thermalization steps in the actual implementation. Thus, the experimental results are collected by simply considering independent stages that require: the preparation of different initial states of the two level system, undergoing some unitary evolution and ultimately carrying out a projective measurement of its state. The article is well written, and the main issues touched upon are timely and of interest to a broad scientific community.

We thank the referee for taking the time to go through our manuscript and for their positive assessment of our work. In particular, we are glad to read that they consider our article "well written", as well as "the main issues touched upon are timely and of interest to a broad scientific community".

However, given that the main contribution of the paper is the experimental confirmation of a quantum correction proposed in Ref. [43,44], my only concern is related to the relevance of the experiment that is presented. After reading the manuscript my impression is that this seems a standard procedure in this particular experimental platform, but there could be some crucial details or elements which I've overlooked or missed entirely. Therefore, I am willing to accept the article for publication, provided the authors present a compelling argument as to why this should be published in such a high-impact journal.

Here, we would like to politely point out that we are convinced that our work is far from being a mere experimental verification of recently theoretical predicted violations to the work-FDR. We are more ambitious than that. In fact, this is a point that is very important to us. We have also taken steps to make this much clearer in the manuscript and sincerely hope that our core message has become manifest.

The first reason comes from the main goal of our work, which is to show how recent results in stochastic thermodynamics can be exploited in order to provide clear-cut and measurable certifications of genuine quantum signatures by means of simple energy measurements in the σ_z basis of a qubit. The simplicity in the measurement scheme is a very desirable feature since certifying genuine quantum properties might entail complicated measurement procedures.

The other, equally important motivation, is that our paper provides important thermodynamic predictions **beyond** the regimes and approximations where theoretical predictions have been made. While a quantum correction to the fluctuation-dissipation relation (FDR) has theoretically been proven in Refs. [43-44] within the slow-driving regime (i.e., in the limit $N \gg 1$ of large number of quench+thermalization steps), existing physical constraints on our state-of-the-art trapped-ion platform allowed us to investigate such violations beyond this regime and for finite - and even fast - processes (i.e., small and intermediate number of steps N).

Our motivation is hence reminiscent of that of quantum state tomography in quantum information science, or even in tests checking for a violation of Bell's inequalities: We are not making a prediction and checking whether the experiment is kind of close to them. Instead, we ask the stronger question to what extent the data obtained in the experiment are compatible with incoherent dynamics.

Furthermore, once again motivated by the inevitable presence of SPAM errors in our experiment, we considered for the first time the impact of SPAM error on the out-of-equilibrium work statistics, ultimately leading to the discovery of a purely SPAM-generated violation of the FDR that had not been theoretically predicted before.

The fact that both these regimes represented an uncharted territory from a theoretical point of view was among the reasons why we included a thorough analysis based on a large number of simulations aimed at statistically excluding that the observed FDR-violations could have had a non-quantum origin, i.e., either due to finite-speed classical incoherent processes or SPAM errors.

To summarize, we do not merely report on an experimental verification of a known theoretical result: on one hand, it aims at showcasing an interesting, novel and potentially very useful application of recent results in quantum stochastic thermodynamics, namely an easily measurable way to certify genuine quantum properties. At the same time, it has made full use of both the high-level of control on trapped ions and of the inevitable experimental constraints in order to provide new results beyond the regimes of theoretical predictions, thus providing essential indications for further developments.

In the resubmitted version of the manuscript, we have restructured the presentation and discussion in order to highlight and better stress all these points. We hope that, in light of this, the Referee will recommend this paper for publication in Nature Communications.

Reply to Reviewer #3

The authors describe an experimental scheme of a trapped ion qubit to verify the quantum work fluctuation-dissipation relation. They performed a sequence of two-time work measurements with a coherent rotational operation in between the measurements. The state preparation and measurement errors are used to deduce the standard deviation, the correction to the work fluctuation-dissipation relation. The paper is well written but would benefit from recapping the derivation of the fluctuation relation. Although the experimental study of fluctuation-dissipation is interesting to the on-going effort to the nonequilibrium phenomena in the information quantum thermodynamics, the present study appears to miss the measurement contribution part. In addition, it is not clear how the violation of the fluctuation-dissipation relation at high temperature is related to the 'genuine' quantum correction. And how does the study differ from other trapped-ion studies on the verification of Jarzynski equality and Landauer principles. In summary, I cannot recommend the article for publication in Nature Communication in its current form.

We would to thank the referee for their reading of our work and for the report. We are happy to read about the assessment of our work being "well-written". We still think, however, that a misunderstanding has occurred concerning what we would like to achieve with our work. With this reply letter, we explain that, but we also lay out in detail what we have done to avoid this misunderstanding. To start with, we would like to firmly and better argue on the marked differences that set it apart from all previous experimental reports on quantum thermodynamics, both in scope and conclusions.

First of all, we emphasize that we are very well aware of the literature about trapped-ion studies on verification of Jarzynski equality (our cited Ref. [47]) and Landauer's principle (our Ref. [50] and the newly added Phys. Rev. Lett. 120, 210601); this stream of experimental research on quantum thermodynamics using trapped-ions is indeed nothing but a strong evidence of the suitability of this type of platform for this kind of research, due to the high-level of control reachable that allows for precise manipulations and measurements.

Any analogy between our work and the afore-mentioned literature, however, stops at this high level of experimental methodology (i.e., trapped-ions) and field of research (quantum thermodynamics). We take in fact the baton of these works and present results that go significantly beyond them both in scope and predictions. In particular, two crucial arguments represent a solid proof of our statement.

First of all, the goal of our work is not to investigate "whether there are violations of the work fluctuation-dissipation relation" (FDR from now on, for brevity). We would have agreed with the referee that this is interesting, but not very interesting, after all. After all, if a relation has been shown to be true, it is not so important to experimentally verify it, as it is anyway true already (provided that quantum mechanics is correct).

In our work, we are more ambitious: We demonstrate the potential of non-equilibrium thermodynamics in unambiguously certifying genuine quantum signatures by showcasing, for the first time, the actual experimental measurability of FDR violations (once practical constraints and errors are included, as commented also below).

Even more importantly, all the *theoretical* predictions regarding the quantum violations of the FDR were proven only within the slow driving regime which, for discretized protocols consisting of N alternating steps of Hamiltonian quenches and thermalizations, corresponds to the limit $N \rightarrow \infty$. Experimentally accessing the slow-driving regime would require being able to perform coherent and precise rotations of infinitesimal angles $\propto 1/N$, which is however currently still out of reach even for state-of-the-art platforms like ours despite the high single qubit fidelities and precision in qubit manipulations.

This practical series of constraints led us to experimentally investigate the violations to the work FDR **beyond** the slow-driving regime, namely for finite and relatively small values of N (i.e., corresponding to finite-time and fast driving protocols), where **no theoretical predictions have been made yet**. The results of our work, therefore, provide valuable information for future advancements in the still uncharted territory of quantum work FDR for finite-time out-of-equilibrium processes. This is also the reason why in our work, unlike e.g. Ref. [47] mentioned by the referee, we did not stop at showing the compatibility of the measured data with a finite-speed extrapolated value of the quantum-friction violations of the FDR, but we performed a thorough analysis, based on a large number of simulations, aimed at statistically ruling out the possibility that the measured violations could be due to finite-speed classical (i.e., incoherent) processes (blue region in Fig. 2).

“Although the experimental study of fluctuation-dissipation is interesting to the on-going effort to the nonequilibrium phenomena in the information quantum thermodynamics, the present study appears to miss the measurement contribution part.”

We must admit that we are not quite sure what the referee precisely means with this claim, since the definition of work we adopt (which is the standard one based on the famous and well-established two-point measurement scheme) is entirely based on identifying the work with the difference of outcomes in energy measurements of the qubit before and after each Hamiltonian quench. The role of measurements is therefore naturally incorporated both in the theoretical framework we consider as well as in the protocol we carried out in the laboratory, as clearly explained in detail on page 2, right column. To make this point even more clear, and again to avoid any misunderstandings, we rewrote this section in order to put even more emphasis on the role of measurements in the definition of the work, which is the central quantity investigated here.

Even more importantly, our research has considered and analyzed *for the first time*, both theoretically and experimentally, the role and impact of SPAM (state preparation and measurement readout) error on the violation of the work FDR. This, in particular, shows that the contribution and effect of measurements on the work statistics to the highest possible degree within the context considered in this work (i.e., the regime of non-equilibrium thermodynamics of a driven closed quantum system).

Again, we thank the reviewer for the report. It has helped us sharpen our main scientific point. We hope that the referee, in light of all these considerations and of the revision done on the manuscript in order to clarify these points better, will revise their decision and reconsider our paper for publication in Nature Communications.

REVIEWER COMMENTS

Reviewer #1 (Remarks to the Author):

In the revised version of the manuscript authors addressed most of my technical comments and clarified the motivations behind this experiments and hypothesis it aims to test. As far I am concerned, the manuscript now can be published in Nature Communications.

Reviewer #3 (Remarks to the Author):

The authors have presented a revised version addressing some of the concerns raised during the initial review process. They have provided clarification regarding the two-point measurement protocol employed in their analysis. However, upon re-evaluation, I find it difficult to discern the novelty of the results presented in the manuscript that would warrant publication for general audience of Nature Communications.

Specific comments:

1. Although they have provided insight into the experimental methodology regarding the two-point measurement protocol, it does not significantly impact the overall assessment of the manuscript's novelty and significance. For instance, how does the 'quantum friction' relates to the energetic of the external control?
2. The manuscript describes the potential occurrence of experimental measurement readout errors, which stems from state preparation and measurement errors. However, further analysis regarding the impact of the environmental or control noise errors on the results and their interpretation would strengthen the manuscript.

Based on the assessment of the revised manuscript, I cannot recommend it for publication in Nature Communications. I encourage the authors to consider exploring alternative publication avenue (subject specific journal) for their research.

Reviewer #4 (Remarks to the Author):

The present paper shows an experimental verification of some prediction of quantum thermodynamics with a trapped ion. In particular, it addresses the question of experimentally finding the quantum contribution to fluctuation dissipation theorem, showing the presence of a quantum correction to the variance of work.

I believe this is an original and significant experimental work, as it undoubtedly establishes the presence of this correction, in a controlled quantum system.

Specifically, I do not share the doubts of previous referees. I think that the results (showing the presence of non-classical features beyond the slow driving regime) are of importance in the realm of quantum thermodynamics.

Specifically, the authors not only provide experimental verification of existing theory (about the deviation of the variance in the slow driving regime), but they go on to analyse their system outside this regime, and they also describe the relevant theoretical analysis.

Concerning the data reported in Fig. 3, I'm not fully convinced of what the authors themselves state in the response to referee 1 (point 6): some more effort should be made to analyse possible error source (even qualitatively).

After this is properly discussed, I would feel quite comfortable with suggesting the acceptance of this manuscript.

Reviewer #5 (Remarks to the Author):

In this work the authors use a single trapped ion as a test-bed to measure a quantum correction to the work fluctuation-dissipation relation (FDR).

The authors perform a two-point measurement (TPM) scheme on a single qubit subjected to a quenching hamiltonian, such that a finite 'rotation' of the Hamiltonian is achieved in N steps.

The system is allowed to thermalize between consecutive unitary operations.

With this scheme, the authors are able to reconstruct the probability distribution of work. From the comparison between the mean work and the work, the authors estimate the correction of the FDR.

Moreover, a detailed analysis allows the authors to rule out the effect of thermal fluctuations and measurement induced errors in the correction factor of the FDR.

In my opinion this work is well written, the results are clearly presented and the discussion is easy to follow.

The results are also timely, since there is a large interest in the development of new experimental protocols to find true quantum signatures in (stochastic) thermodynamic processes.

Therefore, I recommend publication of the manuscript.

I only have one comment regarding the comparison between the coherent and incoherent processes:

Comment:

The study carried out by the authors regarding the measurement errors seems to me very complete and pertinent. In contrast, it is not clear to me how the coherent and incoherent processes are comparable. Even when the correction to the FDR is scaled by the speed v , the two processes seem to me incomparable.

In the incoherent process the energy of the system changes from $\omega_q^{(i)}$ to $\omega_q^{(f)}$. How are the values of $\omega_q^{(f)}$ chosen? Maybe the authors didn't mention some relation that made these two processes comparable?

I'm probably missing something, but from the text I understood that in the incoherent process the Hamiltonian is proportional to σ_z at all times. But then the flip probability p_f (as defined in the supplemental material) would always be zero.

I assume that $\omega_q^{(f)}$ is defined in terms of the angle $\Delta\theta$ as a projection on the Hamiltonian (Z) axis. Such that, after the thermalization in each step, the system is in the same state for the coherent and incoherent process.

I think that a brief explanation on the incoherent process might be very useful for the

reader. In addition, it would also be interesting if the authors comment on the comparison of $\text{var}[W]$, mean W and ΔF for the coherent and incoherent processes. This could be useful to gain more insight on the origins of the variation in the quantum correction to the FDR.

Reply to Reviewer #1

In the revised version of the manuscript authors addressed most of my technical comments and clarified the motivations behind this experiments and hypothesis it aims to test. As far I am concerned, the manuscript now can be published in Nature Communications.

We sincerely thank the Referee for their strong support towards publication in Nature Communications.

Reply to Reviewer #3

The authors have presented a revised version addressing some of the concerns raised during the initial review process. They have provided clarification regarding the two-point measurement protocol employed in their analysis. However, upon re-evaluation, I find it difficult to discern the novelty of the results presented in the manuscript that would warrant publication for general audience of Nature Communications.

We thank the Referee for taking their time to go through our revised version of the manuscript. We are pleased to read that some of the Referee's initial concerns were clarified.

Specific comments: Although they have provided insight into the experimental methodology regarding the two-point measurement protocol, it does not significantly impact the overall assessment of the manuscript's novelty and significance. For instance, how does the 'quantum friction' relates to the energetic of the external control?

We thank the Referee for their question. Since it consists of two different aspects, we believe that a clarification here is in order. On the one hand, the two-point measurement approach is nothing but a standard and well-established methodology to access the statistics of thermodynamic quantities, such as average work and its variance, in quantum systems. Although we are glad to read that our revisions and clarifications had provided further insights on this technique, we never claimed that this was among the novelty elements in our paper, but rather just a necessary tool to define the quantities of interest. On the other hand, as extensively argued and explained both in the paper and in the previous round of review, the novelty and significance in our paper lies in the fact that, for the first time, we provide a unambiguous experimental demonstration that non-equilibrium thermodynamics, and in particular "quantum friction", can be used to certify genuine quantum signatures in a process. Furthermore, our work is the first one to neatly demonstrate this experimentally in a clear setting and even beyond the regimes of validity of the theory.

This brings us to answering the second part of the question: "how does the 'quantum friction' relate to the energetic of the external control". The answer comes from the realisation that the quantity we measure is precisely the work dissipated by the external control in order to drive the process: since the correction term to the fluctuation-

dissipation relation induced by quantum friction is always non-negative, $\mathcal{Q} \geq 0$ for every protocol time τ and every temperature β , this implies that the generation of instantaneous coherence in the energy eigenbasis (i.e. quantum friction) inevitably comes at a higher energetic cost both on average and at the level of the fluctuations than for incoherent processes. To be specific, we have included a detailed analysis that quantitatively compares the average dissipated work and its variance for coherent and for incoherent processes (see Subsection I.A). This allows to conclude that generating quantum friction along a process always entails more work done irreversibly on the system.

The manuscript describes the potential occurrence of experimental measurement readout errors, which stems from state preparation and measurement errors. However, further analysis regarding the impact of the environmental or control noise errors on the results and their interpretation would strengthen the manuscript.

Owing to the exquisite degree of control over single trapped ions, as well as to the almost perfect isolation of the Pauli trap from environmental noise (which was among the reasons why one typically has to emulate the qubit thermalization by incomplete optical pumping, see also Ref. 20 among many others), both of the above-mentioned additional sources of error are negligible compared to the SPAM errors that we have characterised. Previously in Applied Physics B, 122(10), 254, we have measured for the exact same setup the coherence time of the spin states superposition to be approximately 2.1 seconds; this is way longer than any other timescale associated with the operations carried out in the present protocol, meaning that the qubit remains fully coherent (and thus isolated from the decoherence and noise induced by an external environment) throughout the implementation of our experiment. We have included a sentence in the main paper that explicitly clarifies this point.

Furthermore, the temperature is also carefully calibrated and determined using optical pumping rates between spin states, see Phys. Rev. Lett. 123, 080602. For example the Referee can have a look at Fig 2 of Phys. Rev. Lett. 123, 080602 for details and we follow similar techniques in our present work. (sub-percent accuracy of the lifetime which translates here on accuracy of the spin imbalance). Finally, control errors affecting the rotation angles were carefully taken into account: a detailed quantitative discussion concerning this point could be however already found in Section C of the Supplementary Notes.

Based on the assessment of the revised manuscript, I cannot recommend it for publication in Nature Communications. I encourage the authors to consider exploring alternative publication avenue (subject specific journal) for their research.

We thank again the Referee for their assessment and feedback. While we are glad to read that all the other Referees strongly acknowledged both the novelty and significance of our work, we hope that, in light of these additional clarifications, this Referee can also revert their decision on our manuscript.

Reviewer #4 (Remarks to the Author):

The present paper shows an experimental verification of some prediction of quantum thermodynamics with a trapped ion. In particular, it addresses the question of experimentally finding the quantum contribution to fluctuation dissipation theorem, showing the presence of a quantum correction to the variance of work. I believe this is an original and significant experimental work, as it undoubtedly establishes the presence of this correction, in a controlled quantum system. Specifically, I do not share the doubts of previous referees. I think that the results (showing the presence of non-classical features beyond the slow driving regime) are of importance in the realm of quantum thermodynamics. Specifically, the authors not only provide experimental verification of existing theory (about the deviation of the variance in the slow driving regime), but they go on to analyse their system outside this regime, and they also describe the relevant theoretical analysis.

We sincerely thank the Referee for their very strong support of our work.

Concerning the data reported in Fig. 3, I'm not fully convinced of what the authors themselves state in the response to referee 1 (point 6): some more effort should be made to analyse possible error source (even qualitatively).

We thank the Referee for highlighting this important point. Due to the setup and the platform considered, as well as the type of process performed, the other two possible sources of error beside state preparation and measurement readout (SPAM) are: (i) uncertainties in the control operations (i.e. the rotations) and (ii) external noise.

Concerning (i), a large number of subdivisions N for the rotation R [Eq.(5)] with high accuracy is challenging, as the angle becomes extremely small and thus prone to errors. This was the reason why we could not obtain reliable experimental points for higher values of N ; however, such technical constraints pushed us to analyse the finite-speed regime, arriving at results that therefore go beyond the regimes of theoretical predictions. A detailed quantitative analysis of this source of errors was carefully included in the determination of the errors bars in all the plots and discussed in Section C of the Supplementary Notes.

For what concerns external noise (ii), this is negligible in our experiment. This is due to the high degree of control over single trapped ions, and to the almost perfect isolation of the Pauli trap from external environmental noise (which was among the reasons why one typically has to emulate the qubit thermalization by incomplete optical pumping, see also Ref. 20 add more). Previously in *Applied Physics B*, 122(10), 254, some of the Authors measured (for the exact same setup) the coherence time of the spin states superposition, finding it to be equal to 2.1 seconds. Since this is way longer than any other timescale associated with the operations carried out in the present protocol, this means that the qubit remains fully coherent (and thus isolated from the decoherence and noise induced by an external environment) throughout the implementation of our experiment.

Finally, we would also like to emphasise that the temperature is also carefully calibrated and determined using optical pumping rates between spin states, see *Phys. Rev. Lett.* 123, 080602. For example the Referee can have a look at Fig 2 of *Phys. Rev.*

Lett. 123, 080602 for details and we follow similar techniques in our present work. (sub-percent accuracy of the lifetime which translates here on accuracy of the spin imbalance). In conclusion, we agree with the Referee that these considerations were missing and we therefore included them in the revised version.

After this is properly discussed, I would feel quite comfortable with suggesting the acceptance of this manuscript.

We thank once again the Referee for taking their time to go through the manuscript and the other Referee reports in detail, and for their support towards publication in Nature Communications.

Reviewer #5 (Remarks to the Author):

In this work the authors use a single trapped ion as a test-bed to measure a quantum correction to the work fluctuation-dissipation relation (FDR). The authors perform a two-point measurement (TPM) scheme on a single qubit subjected to a quenching hamiltonian, such that a finite 'rotation' of the Hamiltonian is achieved in N steps. The system is allowed to thermalize between consecutive unitary operations. With this scheme, the authors are able to reconstruct the probability distribution of work. From the comparison between the mean work and the work, the authors estimate the correction of the FDR. Moreover, a detailed analysis allows the authors to rule out the effect of thermal fluctuations and measurement induced errors in the correction factor of the FDR. In my opinion this work is well written, the results are clearly presented and the discussion is easy to follow. The results are also timely, since there is a large interest in the development of new experimental protocols to find true quantum signatures in (stochastic) thermodynamic processes. Therefore, I recommend publication of the manuscript.

We sincerely thank the Referee for their careful and detailed read of our manuscript and their strong support towards publication in Nature Communications.

I only have one comment regarding the comparison between the coherent and incoherent processes: Comment: The study carried out by the authors regarding the measurement errors seems to me very complete and pertinent. In contrast, it is not clear to me how the coherent and incoherent processes are comparable. Even when the correction to the FDR is scaled by the speed v , the two processes seem to me incomparable. In the incoherent process the energy of the system changes from $\omega_q^{(i)}$ to $\omega_q^{(f)}$. How are the values of $\omega_q^{(f)}$ chosen? Maybe the authors didn't mention some relation that made these two processes comparable? I'm probably missing something, but from the text I understood that in the incoherent process the Hamiltonian is proportional to σ_z at all times. But then the flip probability p_f (as defined in the supplemental material) would always be zero. I assume that $\omega_q^{(f)}$ is defined in terms of the angle $\Delta\theta$ as a projection on the Hamiltonian (Z) axis. Such that, after the thermalization in each step, the system is in the

same state for the coherent and incoherent process. I think that a brief explanation on the incoherent process might be very useful for the reader.

We thank the Referee for their important observation. We agree that we the previous explanation of the incoherent process was insufficiently clear and we apologise if this generated some confusion concerning how to compute the related work statistics. The answer is that Eqs. (7) - (9) of the main text (or equivalently Eqs. (1) - (3) of the Supplementary Notes) are the particular expressions of the more general work probability distribution from the TPM approach Eq. (6) for coherent processes, i.e. for the one implemented in the experiment and discussed in detail in the Section above these equations of the main text. In order to compute the corresponding expressions for the step work statistics along the incoherent process, one needs to restart from Eq. (6) and will obtain two new simpler expressions. We once again apologise if these expressions were missing and we have now added a whole new Subsection (Subsec. I.A, highlighted in blue color) in the Supplementary Notes where we explain in detail how the incoherent process is calculated and provide the explicit expressions for the above-mentioned probabilities.

In addition, it would also be interesting if the authors comment on the comparison of $\text{var}[W]$, mean W and ΔF for the coherent and incoherent processes. This could be useful to gain more insight on the origins of the variation in the quantum correction to the FDR.

We hope that the newly added Subsection I A. in the Supplementary Notes addresses in details this point. In it, we explain in detail how to derive each quantity for incoherent and coherent processes. From these expressions it can be seen that \mathcal{Q} is non-zero only for coherent processes (at order $1/N$), which enables us to find a genuinely quantum signature (see also Ref. [57] in the main text).

We would like to once again thank the Referee for their insightful and constructive comments which have led us to improve on the quality of our work.

REVIEWERS' COMMENTS

Reviewer #4 (Remarks to the Author):

I'm perfectly satisfied by the author's comments to Fig. 3 and in particular to the errors reported there, which was my only concern in the previous round. Therefore, my advise is that the manuscript deserves publication in Nature Communications.

Reviewer #5 (Remarks to the Author):

The authors have given a satisfactory response to my comments. I suggest publication.